# Reduction in Oviposition of Poultry Red Mite (*Dermanyssus gallinae*) in Hens Vaccinated with Recombinant Akirin

**DOI:** 10.3390/vaccines7030121

**Published:** 2019-09-19

**Authors:** Jose Francisco Lima-Barbero, Marinela Contreras, Kathryn Bartley, Daniel R. G. Price, Francesca Nunn, Marta Sanchez-Sanchez, Eduardo Prado, Ursula Höfle, Margarita Villar, Alasdair J. Nisbet, José de la Fuente

**Affiliations:** 1SaBio, Instituto de Investigación en Recursos Cinegéticos IREC (CSIC-UCLM-JCCM), Ronda de Toledo 12, 13071 Ciudad Real, Spain; josefranvet@gmail.com (J.F.L.-B.); marinelacr@hotmail.com (M.C.); marta.sanchez@uclm.es (M.S.-S.); ursula.hofle@uclm.es (U.H.); margaritam.villar@uclm.es (M.V.); 2Sabiotec, Ed. Polivalente UCLM, Camino de Moledores s/n, 13005 Ciudad Real, Spain; 3Moredun Research Institute, Pentlands Science Park, Bush Loan, Edinburgh, Midlothian EH26 0PZ, UK; kathryn.bartley@moredun.ac.uk (K.B.); dan.price@moredun.ac.uk (D.R.G.P.); francesca.nunn@moredun.ac.uk (F.N.); alasdair.nisbet@moredun.ac.uk (A.J.N.); 4Department of Applied Physics, Faculty of Science, University of Castilla La Mancha, Avda. Camilo José Cela 10, 13071 Ciudad Real, Spain; eduardo.prado@uclm.es; 5Department of Veterinary Pathobiology, Center for Veterinary Health Sciences, Oklahoma State University, Stillwater, OK 74078, USA

**Keywords:** Dermanyssus, Akirin, vaccine, control, poultry red mite, tick, Subolesin

## Abstract

The poultry red mite (PRM), *Dermanyssus*
*gallinae*, is a hematophagous ectoparasite of birds with worldwide distribution that causes economic losses in the egg-production sector of the poultry industry. Traditional control methods, mainly based on acaricides, have been only partially successful, and new vaccine-based interventions are required for the control of PRM. Vaccination with insect Akirin (AKR) and its homolog in ticks, Subolesin (SUB), have shown protective efficacy for the control of ectoparasite infestations and pathogen infection/transmission. The aim of this study was the identification of the akr gene from *D. gallinae* (*Deg-akr*), the production of the recombinant Deg-AKR protein, and evaluation of its efficacy as a vaccine candidate for the control of PRM. The anti-Deg-AKR serum IgY antibodies in hen sera and egg yolk were higher in vaccinated than control animals throughout the experiment. The results demonstrated the efficacy of the vaccination with Deg-AKR for the control of PRM by reducing mite oviposition by 42% following feeding on vaccinated hens. A negative correlation between the levels of serum anti-Deg-AKR IgY and mite oviposition was obtained. These results support Deg-AKR as a candidate protective antigen for the control of PRM population growth.

## 1. Introduction

The poultry red mite (PRM), *Dermanyssus gallinae* (De Geer, 1778), is a hematophagous parasitic mite of birds. It has a worldwide distribution and is considered the major pest for the poultry industry in Europe and Asia with severe economic loses for the egg-production sector [1,2]. The PRM hides in furniture in the hen’s surroundings and feeds on the hens in darkness [3,4]. The lifecycle of PRM includes five stages: egg, larvae, protonymph, deutonymph, and adult, and it is usually completed in 2 weeks [5]. Severe infestations by PRM increases hen mortality by provoking severe anemia and behavioral disorders [6,7,8]. PRM has also been shown to be a vector for multiple pathogenic viruses and bacteria [2,9,10].

The control of PRM is primarily based on the use of acaricides but there are a number of associated drawbacks such as the selection of resistant mite populations, environmental/food contamination, and limited success once the infestation is established [2,11,12]. Alternative control methods against PRM are under development, and vaccination is a promising intervention as it is environmentally sound, reduces the use of acaricides, and avoids the selection of resistant mites. The use of recombinant proteins for vaccinating hens against PRM has shown favorable results under in-vitro mite-feeding conditions [13,14,15,16].

Akirin (AKR) is a protein encoded by highly-conserved *akirin* (*akr*) genes involved in different biological processes, including the innate immune response in all metazoans [17]. The AKR ortholog in ticks, Subolesin (SUB, also known as 4D8), is involved in several biological processes such as tick response to infection, feeding, stress response, development, and reproduction [17]. Most vertebrates have two AKR homologues (AKR1 and AKR2) while insects and ticks have only one SUB/AKR [18]. The SUB/AKR orthologues have shown vaccine efficacy for the reduction of ectoparasite infestations and pathogen infection/transmission [17,19]. Recombinant AKR from *Aedes albopictus* has been previously evaluated as a vaccine candidate against PRM, resulting in statistically significantly higher mite mortality when fed on blood from vaccinated hens when compared to control birds [15]. However, one limitation of this previous study is the use of mosquito AKR as the PRM homologue sequence was unavailable at that time. The mosquito AKR may therefore have limited protection efficacy against PRM in vaccinated hens when compared to endogenous AKR antigen.

To address this limitation, in this study we identified the *akr* gene sequence from *D. gallinae* (Deg-akr) from the newly-available draft genome resource [20]. The recombinant Deg-AKR protein was then produced in Escherichia coli and used to evaluate its efficacy as a vaccine candidate for the control of PRM using a novel on-hen feeding device [21].

## 2. Materials and Methods

### 2.1. Ethics Statement

Animal experiments were conducted in strict accordance with the guidelines of the European Community Directive 2010/63/EU. Animals were housed in the experimental farm of the Institute for Game and Wildlife Research (IREC) with the approval and supervision of the Ethics Committee on Animal Experimentation of the University of Castilla La Mancha (Registry number PR-2018-11-20).

### 2.2. Mites

*Dermanyssus gallinae* of mixed developmental stages and sexes were obtained from a commercial egg-laying farm in Consuegra (Toledo, Spain). Mites were stored in vented 75 cm^2^ tissue culture flasks (Corning, NY, USA) at room temperature (RT) for 10 days. Adult females were selected based on size and morphology, and protonymphs were obtained from larvae hatched from previously-harvested mite eggs.

### 2.3. Cloning of the Gene Coding for Deg-AKR

In order to identify the *Deg-akr* gene in PRM, the AKR amino acid sequence from *Metaseiulus occidentalis* (XM_003738959.2) was used as a reference sequence in a local tBLASTn search against the draft genome of the PRM with GenBank accession number QVRM01000000 [20]. Based on sequence similarity to the *M. occidentalis* AKR gene, exons encoding the *Deg-akr* ortholog were identified in the *D. gallinae* genome, and the full-length coding sequence (CDS) was manually assembled. The full-length CDS was PCR-amplified from *D. gallinae* cDNA generated from mixed-population *D. gallinae* mites collected from a commercial egg-laying facility (Scotland, UK) as described previously [22]. PCR amplification of the *Deg-akr* CDS was performed using the oligonucleotide primers Sub_F1: 5′-ATGGCATGTGCGACGCTAAAACGTC-3′ and Sub_F2: 5′-TTACGAGCAATAGGACGGGGCGC-3´ that were designed to amplify the full CDS including the putative initiating methionine and stop codon that were conserved between different species of the Acari (Figure 1). PCR amplification was completed using proof-reading DNA polymerase and performed on an Applied Biosystems 2720 thermal cycler with the following conditions: 94 °C for 2 min, followed by 30 cycles at 94 °C for 30 s, 61 °C for 30 s and 72 °C for 1 min. A multiple amino acid sequence alignment was performed using the Clustal Omega algorithm [23] to identify conserved regions. The amplification products were ligated into the ChampionTM pET SUMO vector (Thermo Fisher Scientific, Waltham, MA, USA) and confirmed by DNA sequencing (Eurofin Genomics, Luxemburg). The sequence of *Deg-akr* was submitted to GenBank under accession number MN310557.

### 2.4. Production of Recombinant Deg-AKR and Vaccine Formulation

The *Deg-akr* coding sequence was sub-cloned into ChampionTM pET101 expression vector (Thermo Fisher Scientific) and used to transform BL21 StarTM (DE3) E. coli cells. For Deg-AKR protein production, cells were grown at 37 °C with shaking at 200 rpm until OD_600_ = 0.6, and then cultures were induced with 1 mM isopropyl β-D-1-thiogalactopyranoside (IPTG) and cultured for a further 4.5 h. Recombinant Deg-AKR was purified from insoluble cell-lysates in the presence of 7 M urea by Ni affinity chromatography (Genscript Corporation, Piscataway, NJ, USA) as previously described [21] using 1 mL HisTrap FF columns mounted on an AKTA-FPLC system (GE Healthcare, Piscataway, NJ, USA). Purified recombinant Deg-AKR was refolded by dialysis against 1000 volumes of PBS (137 mM NaCl, 2.7 mM KCl, 10 mM Na2HPO4, 1.8 mM KH2PO4), pH 7.4 for 12 h at 4 °C. For vaccine formulation, recombinant protein or PBS saline control were adjuvanted in MontanideTM ISA 71 VG (SEPPIC, Paris, France) as previously described [25]. The final concentration for Deg-AKR was 100 µg/mL in a vaccination dose volume of 0.5 mL.

### 2.5. Hen Vaccination and PRM Infestation

Five 25-week-old Lohman Brown hens per group were randomly assigned to two groups (Group 1 and Group 2) and housed together. Hens were vaccinated on days 0 (V1), 14 (V2) and 28 (V3) with a 0.5-mL intramuscular injection in the breast muscle. Group 1 birds were vaccinated with 50-µg Deg-AKR formulated in MontanideTM ISA 71 VG, and Group 2 birds were vaccinated with an equivalent volume of PBS in MontanideTM ISA 71 VG. PRM exposure was carried out using an on-hen feeding device as described previously [22]. Briefly, a feeding device was placed onto each thigh that had been previously plucked, attached to the skin with Leucoplast tape (BSN Medical, Hamburg, Germany) and secured with cohesive bandage (Henry Schein Inc., Melville, NY, USA). Each device contained 100 starved female adults and 75 protonymphs. Deuthonymphs were not included in the infestation because they are difficult to differentiate from adult males while preparing the pouches for infestation. Mites fed on hens for 3 h. On-hen feeding experiments were repeated independently four times using fresh mites on each assay on days 42, 44, 46, and 49 from the first vaccination.

### 2.6. Assessment of Vaccine Efficacy

Mites were considered fed when they had taken a fresh blood meal and were fully engorged; and partially fed when they had fresh blood ingested but they were not fully engorged. PRM were considered dead if they did not show any movement and were unresponsive to touch stimulus. The number of fed, unfed and partially fed adults and protonymphs per hen were counted individually in each assay. Both fully engorged and partially fed protonymphs and adult mites were collected from the devices and transferred into individual wells on a 96-well tissue culture plate (Corning Costar, Sigma-Aldrich, St. Louis, MO, USA) and sealed with AeraSeal film (Sigma-Aldrich, A9224-50EA, St. Louis, MO, USA). Plates containing fed mites were placed into an incubator at 25 °C and 85% relative humidity and checked daily for 7 days. Adult mites were monitored for mortality, oviposition and egg fertility, while protonymphs were monitored for mortality and molting (Figure 2). In order to determine differences in engorgement between vaccinated and control groups, a body size of five to six fully engorged adult mites per hen was measured by scanning electron microscopy (SEM). The mites used for SEM photography were dehydrated in absolute ethanol for 24 h. Specimens were mounted onto standard aluminum SEM stubs using conductive carbon adhesive tabs. Mites were observed and photographed with a field emission scanning electron microscope (Zeiss GeminiSEM 500, Oberkochen, Germany) operating in high vacuum mode at an accelerating voltage of 2 kV in the absence of metallic coating. Mite body length and width were measured, and body area calculated using a Fiji ImageJ Software [26]. This procedure does not affect acari body size if they are not damaged, which are the samples selected for analysis.

### 2.7. Analysis of Immune Response to Deg-AKR in Hens by ELISA

Blood samples from each hen were collected before each vaccination dose on days 0, 14 and 28 and before PRM feeding on day 42. After each blood collection, serum was separated by centrifugation and stored at −20 °C. Hen eggs (8–10 eggs per group) were collected during 2 consecutive days on days 15–16 (T1), 29–30 (T2) and 42–43 (T3). During egg collection, experimental groups were housed separately. Antibodies from egg yolk were extracted as previously described [27] and stored at −20 °C. The IgY antibody levels against Deg-AKR and a mite-soluble protein extract obtained as previously described [28] were determined by ELISA in sera and egg yolks. High absorption-capacity 96-well plates (Immunoplate, SPL Life Sciences, Syeonggi-do, Korea) were coated overnight at 4 °C with 0.25 µg/well of each antigen. ELISA plates were washed four times with 200 µL PBST (PBS containing 0.05% v/v Tween-20) and incubated with 200 µL/well of blocking buffer (5% w/v skimmed milk powder in PBS, Condalab, Torrejón de Ardoz, Spain) for 2 h at RT on a shaker. Then, 50 µL of serum samples diluted in PBS to 1/2000 and 50 µL of yolk samples diluted to 1/1800 were added. After incubating for 1 h at RT, the plates were washed four times and then incubated for 1 h at RT with 50 µL/well of rabbit anti-IgY-peroxidase conjugate (Sigma-Aldrich) diluted 1/1000 in PBST. After four washes, the reaction was visualized by adding 100 μL of 3,3, ′5,5-tetramethylbenzidine (TMB) (Promega, Madison, WI, USA) and incubated for 10 min in the dark at RT. After stopping the reaction with 50 μL sulfuric acid, the optical density (OD) was measured at 450 nm using a Multiskan GO microplate reader (Thermo Fisher Scientific, Waltham, MA, USA).

### 2.8. Analysis of Hen Immune Response to Deg-AKR by Western Blot

Western blot analysis was carried out as previously described [29]. Briefly, 10 µg of purified recombinant Deg-AKR was separated by electrophoresis on an SDS-12% polyacrylamide gel (Life Science, Hercules, CA, USA) and then stained with Coomassie Brilliant Blue (Sigma-Aldrich, St. Louis, MO, USA) or transferred to a nitrocellulose membrane. The membrane was blocked with 5% BSA (Sigma-Aldrich) for 2 h at RT and washed four times with TBST (50 mM Tris-Cl, pH 7.5, 150 mM NaCl, 0.5% Tween 20). Pooled sera from the day 42 of blood extraction were used as primary antibodies diluted to 1/1000 in TBS. Rabbit anti-IgY-peroxidase conjugate (Sigma-Aldrich, St. Louis, MO, USA) diluted 1/1000 in TBS with 3% BSA was used as the secondary antibody. The membrane was washed five times with TBST and developed with TMB stabilized substrate for HRP (Promega, Madrid, Spain).

### 2.9. Statistical Analysis

Anti-AKR IgY levels between vaccinated and control groups were compared by Mann-Whitney U test (*p* = 0.05). Partially fed and dead, or missing mites were removed from the analysis of the effect of vaccination on PRM reproduction. Pearson´s correlation coefficient was used to determine if the specific AKR antibody levels at OD450 nm of individual vaccinated and control hens at day 42 (the time when mite infestations were performed) are correlated with the total number of eggs laid per mite fed on individual hens. A generalized linear mixed model (GLMM) was performed to determine the group differences in oviposition based on a Negative Binomial distribution and log as link function. The model included the total number of eggs laid per mite (oviposition) as the dependent variable. The treatment group was included as a fixed effect and hen number and assay replicate as random factors. Data exploration, correlations and GLMM were done using the IBM SPSS Statistics v23 software.

## 3. Results and Discussion

Mosquito AKR was previously used as a vaccine candidate against PRM in an experiment that provided support for the role of AKR as a protective antigen [15]. The aim of our study was to identify and determine the protective capacity of the Deg-AKR for the control of PRM infestations in vaccinated hens. The experimental design included the evaluation of vaccination effect on the PRM life cycle including mite (adult and protonymph) mortality, oviposition, and molting (larvae to protonymph and protonymph to deutonymph) (Figure 2).

Alignment of the amino acid sequences obtained from Deg-AKR with AKR/SUB from mites (M. occidentalis and Varroa destructor), tick (*Ixodes ricinus*) and mosquito (*A. albopictus*) showed the presence of conserved regions in the coding sequence (Figure 1), further supporting the evolutionary conservation of AKR/SUB between different species [17].

After vaccination with Deg-AKR, hens developed an antigen-specific IgY immune response, which was detected in both hen sera (Figure 3A) and egg yolk (Figure 3B). Although IgY antibodies in egg yolk do not affect vaccine efficacy, this approach has been used as a non-invasive method to monitor antibody titers in hens [27]. The anti-Deg-AKR serum IgY antibody levels increased during the entire experiment from day 0 to day 42 and remained higher than in control animals until the end of the experiment (Figure 3A). In the egg yolk, the anti-Deg-AKR IgY levels were also higher in the vaccinated than in the adjuvant-only control group (Figure 3B). The IgY response to the soluble mite protein extracts was also higher in the vaccinated than the control group but the difference was only statistically significantly (*p* < 0.05) higher on days 14 and 28 and days 28 and 42 after vaccination for serum and egg yolk, respectively (Figure 3C,D). The soluble PRM protein extract is a complex protein mix, which may reflect an AKR dilution effect that can explain the differences in the antibody levels against Deg-AKR and PRM protein extract (Figure 3A,B vs. Figure 3C,D). The serum IgY recognition of Deg-AKR (25.9 kDa) in vaccinated hens was also corroborated by Western blot (Figure 3E). Serum antibodies also detected the presence of potential Deg-AKR multimers and some protein degradation products, which commonly occur with AKR and SUB [15,17].

Vaccination against ectoparasites is not solely focused on the prevention of the infestation but also in the reduction of the parasite population [30]. This reduction can be achieved by affecting parasite feeding, reproduction and/or development by antigen-specific antibodies ingested after feeding on vaccinated animals [31]. Vaccination with Deg-AKR resulted in a 42% reduction in mite oviposition (GLMM, F = 6.060, df1 = 1, df2 = 747, *p* = 0.014) (Table 1 and Figure 2). However, in this experiment, partially engorged adult mites did not oviposit and among fed mites only 22.4% (168 out of 749) oviposited. These facts resulted in a lower number of eggs/mite (0.3 to 1.0; Table 1) when compared to previous reports (>3.0) [32,33]. The effect of vaccination with AKR on the oviposition in *D. gallinae* has not been reported before, but SUB/AKR has demonstrated a reduction in the oviposition of several ectoparasite species after feeding on vaccinated hosts [17,32,34,35].

As seen in previous vaccine efficacy experiments with SUB/AKR in ticks and insects [29,31,35,36,37], the anti-Deg-AKR IgY levels present in the sera and egg yolk showed a significant negative correlation (r = −0.106 (sera) and r = −0.125 (yolk), *p* < 0.01) with the amount of eggs laid per fed mite (Figure 4A,B). In ticks, the effect of the vaccination on the oviposition is generally correlated with a reduction in the body size and weight of engorged females [31,32]. However, herein we did not detect variations in the body size of the fully engorged mites that were collected from vaccinated and control hens (Figure 4C,D). The average body length in engorged female mites was 1039 µm (from 830 to 1230 µm), the average body width was 563 µm (from 465 to 657 µm), and the average body area was 463 mm^2^ (from 317 to 633 mm^2^). Nevertheless, although selected mites looked like they were fully engorged when sorted with the stereoscope, the variation in body size suggested different blood volumes ingested during the blood meal. These experiments suggested that the use of SEM is a useful tool for measuring blood meal ingestion in individual organisms or developmental stages that cannot be weighed.

In this study, vaccination with Deg-AKR did not significantly affect other mite developmental stages. No statistically significant effect was observed after vaccination with the Deg-AKR on feeding rates and mite mortality in adults or in protonymphs or development (i.e., molting rates in fed and partially fed protonymphs and larval hatching) (Table 2). All eggs laid by females mites hatched by day 7 without differences between groups. During this experiment, lower feeding rates in adult mites were observed when compared to other studies where the on-hen feeding device was applied [21,25]. In our study, the 19 ± 6 % of the starved adult females fully fed across the four replicates. Of the protonymphs, 9 ± 4% were completely engorged, which is a similar feeding rate to that obtained in previous experiments [21]. Adult mites were starved at RT for 10 days, which is a different conditioning protocol from the suggested 3 weeks of starving at 4 °C [21]. However, we used these mites because they appeared healthier than the ones we starved under the recommended conditions [21]. Of the adult mites recovered, 7 ± 4 % were partially fed, but no differences in the number of partially-fed mites were observed between groups. Vaccination with the mosquito AKR was described to increase mite mortality by 35% after an in-vitro feeding assay [15]. However, in-vitro tests have been shown to present a high background mortality of mites [22]. The in-vitro test employs heparinized blood from vaccinated and control hens and mites feed on it through a natural or artificial membrane in an incubator at 36 °C and 75–85% relative humidity [15]. Therefore, mite mortality effects seen in in-vitro feeding systems may be induced by vaccination alone or exacerbated by the high temperatures required to induce mite feeding in the chamber and the presence of anticoagulants in the blood [22,38]. The use of the on-hen feeding device allows mites to feed directly on the hen which circumvents many of these limitations with the in-vitro feeding device and is more likely to present a true reflection of vaccine effects.

## 4. Conclusions

The results obtained in this study identified the *D. gallinae* AKR homolog and demonstrated the efficacy of the vaccination with the Deg-AKR antigen for the control of PRM in hens. The 42% reduction in mite oviposition generated by the vaccination supports the Deg-AKR as a candidate protective antigen for the control of PRM.

## Figures and Tables

**Figure 1 vaccines-07-00121-f001:**
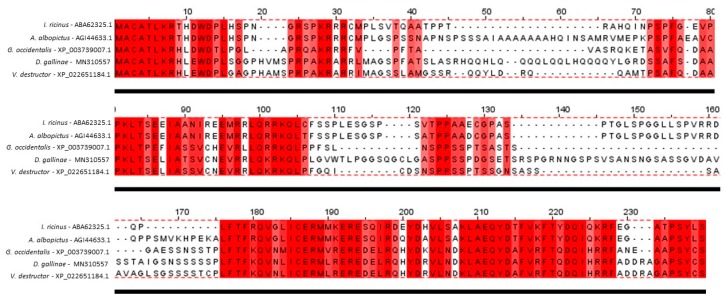
Amino acid protein sequence for *D. gallinae* AKR. Alignment of Deg-AKR amino acid sequence with AKR/SUB sequences from different species. Protein accession numbers are shown. In red are shown regions conserved across different species. The intensity of the red color indicates the level of conservation in that amino acid across the species. Alignment was carried out with Clustal Omega [23] and visualized with Jalview 2.11 software [24].

**Figure 2 vaccines-07-00121-f002:**
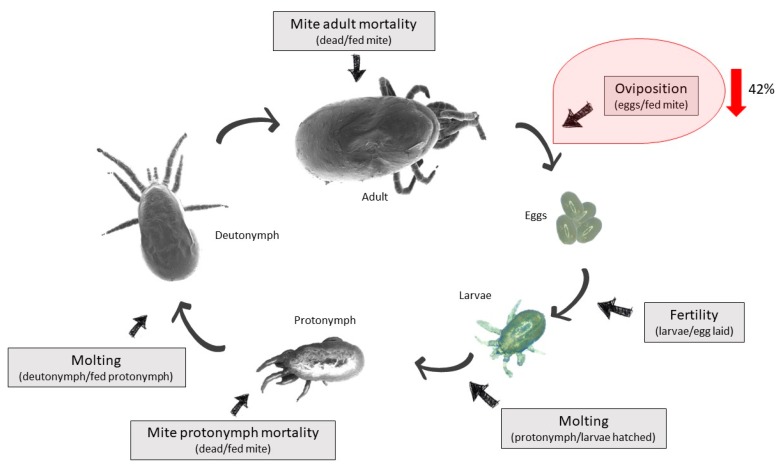
Life cycle of *D. gallinae* and checkpoints for the evaluation of vaccine efficacy. The SEM images for the hematophagous stages of PRM are shown as a representation of the life cycle. During the on-hen feeding assay, the following parameters were recorded per individual hen: adult mite mortality (number of dead/fed mites), oviposition (number of eggs laid/fed mite), fertility (number of larvae/eggs laid), larvae molting (number of protonymphs/larvae hatched), protonymph mortality (number of dead/fed protonymphs), and protonymphs molting (number of deutonymphs/ protonymphs). The results showed a 42% reduction in oviposition (GLMM; F = 6.06, *p* = 0.014, gl1 = 1, gl2 = 747) of mites fed on vaccinated hens when compared to controls.

**Figure 3 vaccines-07-00121-f003:**
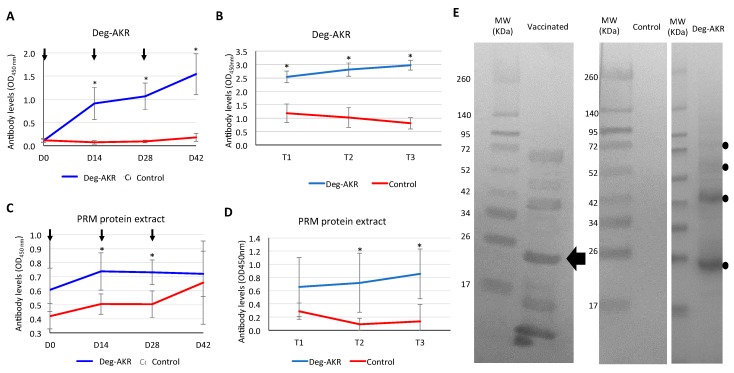
Antibody response to vaccination in hens. Serum samples were collected before each vaccination (arrows) and feeding device application (D42). Antibodies were extracted from eggs collected during two consecutive days on days 15–16, 29–30 and 42–43. Antibody titers are represented as the average OD450nm ± SD and compared between vaccinated and control groups by Mann-Whitney U test (* *p* < 0.05; N = 5). (**A**) Serum IgY antibody titers determined by ELISA against the recombinant Deg-AKR used for vaccination. (**B**) Egg yolk IgY antibodies titers determined by ELISA against the recombinant Deg-AKR used for vaccination. (**C**) Serum IgY antibody titers determined by ELISA against a PRM soluble protein extract. (**D**) Egg yolk IgY antibody titers determined by ELISA against a PRM soluble protein extract. (**E**) Characterization by Western blot of anti-Deg-AKR IgY. Ten micrograms of purified Deg-AKR were used for the analysis of pooled sera collected at D42 from vaccinated and control hens. The position of the recombinant Deg-AKR monomer (25.9 kDa) is marked with an arrow. Other recognized protein bands correspond to potential multimerization or degradation products of the recombinant protein which can also be observed in the Deg-AKR after purification by Ni affinity chromatography (dots). Abbreviation: MW, molecular weight marker (spectra multicolor broad range protein ladder; Thermo Fisher Scientific).

**Figure 4 vaccines-07-00121-f004:**
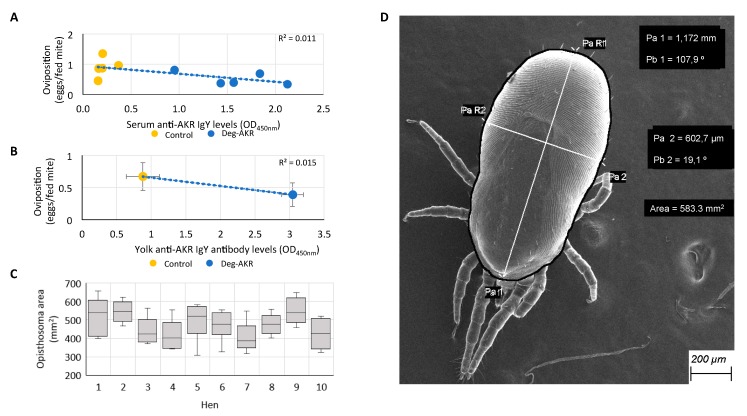
Effect of hen vaccination on PRM oviposition. Antibody levels negatively correlate with the number of eggs laid per fed mite. (**A**) Negative correlation between the levels of serum anti-Deg-AKR IgY and mite oviposition (number of eggs laid per fed adult female mite during the four on-hen feeding tests performed on day 7 of mite monitorization) (r = −0.106, *p* < 0.01). (**B**) Negative correlation between the levels of egg yolk anti-Deg-AKR IgY and mite oviposition (average number of eggs laid per fed adult female mite) (r = −0.125, *p* < 0.01; N = nine egg samples per group with egg collection during 2 consecutive days on days 15–16 (T1), 29–30 (T2) and 42–43 (T3). (**C**) Opisthosoma area in fully engorged adult female mites using five to six mites per hen during the fourth on-hen feeding assay. (**D**) Measures taken for each individual mite to assess the body size after feeding. Mites were photographed with a field emission scanning electron microscope. Mite body length and width were measured, and body area calculated using a Fiji ImageJ Software [23].

**Table 1 vaccines-07-00121-t001:** Summary of adult female PRM feeding rates and vaccine efficacy.

Group	Hen	Fed Mites	Partially Fed/Unfed Mites	Total	% Fed Mites	Average Feeding ± SD	Reduction	Eggs	Oviposition	Average Oviposition ± SD	Reduction	Dead Mites	Mortality	Average Mortality	Mortality Effect
**Deg-AKR**	1	56	38/276 = 314	370	15	21 ± 5	0%	34	0.6	0.4 ± 0.2	42% *	4	0.1	0.2 ± 0.1	6%
	2	77	5/343 = 348	425	18			23	0.3			19	0.2		
	3	130	23/325 = 348	478	27			34	0.3			16	0.1		
	4	120	10/391 = 401	521	23			34	0.3			24	0.2		
	5	81	30/246 = 276	357	23			42	0.5			19	0.2		
**Control**	6	100	23/230 = 253	353	28	18 ± 8		65	0.6	0.7 ± 0.2		15	0.2	0.2 ± 0.0	
	7	32	32/304 = 336	368	9			23	0.7			4	0.1		
	8	59	23/394 = 417	476	12			38	0.6			10	0.2		
	9	85	23/284 = 307	392	22			86	1.0			13	0.2		
	10	44	34/190 = 224	268	16			15	0.3			10	0.2		

Data shown are a compilation from the four assays. Feeding rates were scored on day 1 of the assay, immediately following removal of the feeding devices from the hens. Oviposition and mortality were scored on day 7. Data were analyzed statistically to compare results between mites fed on vaccinated and control hens by GLMM (F = 6.06, *p* = 0.014, gL1 = 1, gL2 = 747). Fed mites = total of fully engorged adult female mites recovered. Unfed mites = total unfed adult female mites were counted after 3 h placed on the hen. Fed reduction = 1–(Average Feeding Deg-AKR/Average Feeding Control)] × 100. Eggs = total accumulative count of eggs laid at day 7. Oviposition = eggs laid/number fed adult female mites. Oviposition reduction = [1–(Average Oviposition Deg-AKR/Average Oviposition Control)] × 100. GLMM: F = 6.060, *p* = 0.014. Dead mites = total number of fed adult female mites which shows no response to stimuli and looks dehydrated at day 7. Mortality = dead mites/number fed adult female mites. Mortality effect = [1–(Average Mortality Deg-AKR/Average Mortality Control)] × 100.

**Table 2 vaccines-07-00121-t002:** Summary of protonymphs PRM feeding and molting rates and vaccine efficacy.

Group	Hen	Fed/PFPN	UnfedPN	Total PN	% Fed	Average Feeding ± SD	Feeding Effect	Molt Fed PN	Molt/Fed	Average Molting ± SD	Molting Effect	Dead PN	Fed	Dead/Fed	Average Mortality	Mortality Effect
**Deg-AKR**	1	13/5 = 18	133	151	11.9	13.1 ± 4.0	0%	5	0.38	0.2 ± 0.1	0%	1	13	0.1	0.2± 0.1	50%
	2	11/4 = 15	101	116	12.9			2	0.18			2	11	0.2		
	3	10/5 = 15	146	161	9.3			1	0.10			3	10	0.3		
	4	33/5 = 38	153	191	19.9			7	0.24			6	33	0.2		
	5	19/1 = 20	157	177	11.3			6	0.32			2	19	0.1		
**Control**	6	4/9 = 13	126	139	9.4	10.6 ± 1.6		0	0.00	0.3 ± 0.3		1	4	0.3	0.1 ± 0.2	
	7	12/1 = 13	87	100	13.0			7	0.58			0	12	0.0		
	8	17/3 = 20	174	194	10.3			3	0.18			1	17	0.1		
	9	15/0 = 15	119	134	11.2			5	0.33			5	15	0.3		
	10	5/7 = 12	120	132	9.1			3	0.60			0	5	0.0		

Data shown are a compilation from the four assays. Feeding rates were scored on day 1 of the assay immediately following removal of the feeding devices from the hens. Molting and mortality were scored on day 7. Data were analyzed statistically to compare results between fed and partially-fed (PF) protonymphs (PN) fed on vaccinated and control hens by a Mann-Whiney test (*p* = 0.05). Fed/PF PN = total of fully engorged/partially fed protonymphs recovered. Unfed PN = total of unfed protnymphs counted after 3 h placed on the hen. Total PN = Fed/PF + Unfed PN. Feeding effect = [1–(Average Feeding Deg-AKR/Average Feeding Control)] × 100. Molt fed PN = total accumulative count of protonymphs which molted to deutonymphs at day 7. Molting effect = [1–(Average Molting Deg-AKR/Average Molting Control)] × 100. Dead PN = total number of fed protonymphs which shows no response to stimuli and look dehydrated at day 7. Mortality = dead protonymphs/number fed protonymphs. Mortality effect = [1–(Average Mortality Deg-AKR/Average Mortality Control)] × 100. Molt PF = total of partially fed protonymphs recovered which molted to deutonymphs. Molting effect PF = [ 1–(Average Molting PF Deg-AKR/Average Molting PF Control)] × 100. Dead PF = total number of partially fed protonymphs which show no response to stimuli and look dehydrated at day 7. Mortality PF = dead partially-fed protonymphs/number of partially fed protonymphs. Mortality effect = [1–(Average Mortality PF Deg-AKR/Average Mortality PF Control)] × 100.

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
