# Peer review of "Reduction in Oviposition of Poultry Red Mite (*Dermanyssus gallinae*) in Hens Vaccinated with Recombinant Akirin"

_vaccines, 2019, doi:10.3390/vaccines7030121_

Round 1
Reviewer 1 Report
PRM infestation is a globally serious threat to the commercial egg-laying industry. Vaccination is a promising alternative control method for currently chemical control methods, which have many drawbacks such as environmental/food contamination, selection of resistant mite populations. The manuscript provides an interesting study in the field of development of vaccine against PRM. The results showed the 42% reducing oviposition rate was obtained when PRMs fed on hens vaccinated with Deg-AKR compared with those in control PRM. This results are very interesting in the development of vaccine against PRMs, which is a challenging work. In my view this study was well designed and conducted. The obtained results are credible, the interpretation of these results is appropriate. The manuscript is very well written. The MS reads well; the style and logical structure, the analysis, discussion of the results are intellectually entertaining.
Some suggestions and comments:
1. Abstract: The results of antibody response to Deg-AKP are major results of the study. Describe them briefly in the abstract.
2. L34: Not only in Europe, but also in other areas, such as in Asia.
3. L73: “2”after“cm” should be superscript.
4. L139-146: Did the preparation of mite sample for SEM have an influence on the size of PRMs? Can the sizes of PRMs be determined under stereoscope since this method has no negative or minor (if it has) effects on the size of mites (L271-273)?
5. L163-164: Why the IgY antibody levels against Deg-AKP in egg yolks were determined since it makes no contribution to the efficacy of Deg-AKP against PRMs and the antibody levels in plasma have been determined?
6. L294-308: Were there partially engorged protonymphs? What’s difference in the oviposition between partially engorged adults and fully engorged ones?
7. Table 1: The average oviposition of PRMs in both groups (0.4 in vaccinated group and 0.7 in control) in the present study is far lower than those previously reported (3.50 observed by Nordenfors et al., 1999 and 3.46 observed by Wang et al., 2018). The possible reasons and the effects of these data on the results as well as on conclusion deserves further discussion.
8. I suggest further discussion on the limits of available efficacy evaluation methods of vaccine against PRMs, including under laboratory and field conditions. This work will help readers to better know why the authors chose this evaluation method in the present study and how to choose suitable evaluation methods in their studies.
I recommend to be published after revision.
Author Response
Reviewer 1
PRM infestation is a globally serious threat to the commercial egg-laying industry. Vaccination is a promising alternative control method for currently chemical control methods, which have many drawbacks such as environmental/food contamination, selection of resistant mite populations. The manuscript provides an interesting study in the field of development of vaccine against PRM. The results showed the 42% reducing oviposition rate was obtained when PRMs fed on hens vaccinated with Deg-AKR compared with those in control PRM. These results are very interesting in the development of vaccine against PRMs, which is a challenging work. In my view this study was well designed and conducted. The obtained results are credible, the interpretation of these results is appropriate. The manuscript is very well written. The MS reads well; the style and logical structure, the analysis, discussion of the results are intellectually entertaining.
Some suggestions and comments:
Abstract: The results of antibody response to Deg-AKP are major results of the study. Describe them briefly in the abstract.
Response: Following reviewer suggestion the Abstract was revised to include that “The anti-Deg-AKR serum IgY antibodies in hen sera and egg yolk were higher in vaccinated than control animals throughout the experiment. The results demonstrated the efficacy of the vaccination with Deg-AKR for the control of PRM by reducing mite oviposition by 42% following feeding on vaccinated hens. A negative correlation between the levels of serum anti-Deg-AKR IgY and mite oviposition was obtained.”
L34: Not only in Europe, but also in other areas, such as in Asia.
Response: Done.
L73: “2”after“cm” should be superscript.
Response: Done.
L139-146: Did the preparation of mite sample for SEM have an influence on the size of PRMs? Can the sizes of PRMs be determined under stereoscope since this method has no negative or minor (if it has) effects on the size of mites (L271-273)?
Response: According to the procedure used for SEM as described in the paper (section 2.6), “The mites used for SEM photography were dehydrated in absolute ethanol for 24 h. Specimens were mounted onto standard aluminum SEM stubs using conductive carbon adhesive tabs.” This procedure does not affect acari body size if they are not damaged, which are the samples selected for analysis. In response to reviewer comment, this information was added to the methods.
L163-164: Why the IgY antibody levels against Deg-AKP in egg yolks were determined since it makes no contribution to the efficacy of Deg-AKP against PRMs and the antibody levels in plasma have been determined?
Response: Antibody levels in egg yolks were determined to provide additional evidences of the antibody response to vaccination. Additionally, as now mentioned in the revised R&D this approach is used as a non-invasive method to monitor antibody titers in hens.
L294-308: Were there partially engorged protonymphs? What’s difference in the oviposition between partially engorged adults and fully engorged ones?
Response: In response to reviewer questions, protonymphs results were included in new Table 2. Results of oviposition in partially and fully engorged adults was included in revised R&D (see response to next comment).
Table 1: The average oviposition of PRMs in both groups (0.4 in vaccinated group and 0.7 in control) in the present study is far lower than those previously reported (3.50 observed by Nordenfors et al., 1999 and 3.46 observed by Wang et al., 2018). The possible reasons and the effects of these data on the results as well as on conclusion deserves further discussion.
Response: The difference between these results is due to the fact that in our experiment not all fed adults oviposited, thus reducing the number of eggs per mite. To address reviewer comment the R&D was revised to include that “However, in this experiment partially engorged adult mites did not oviposit and among fed mites only 22.4% (168 out of 749) oviposited. These facts resulted in lower number of eggs/mite (0.3 to 1.0; Table 1) when compared to previous reports (> 3.0) [32, 33]”
I suggest further discussion on the limits of available efficacy evaluation methods of vaccine against PRMs, including under laboratory and field conditions. This work will help readers to better know why the authors chose this evaluation method in the present study and how to choose suitable evaluation methods in their studies.
I recommend to be published after revision.
Response: Following reviewer suggestion, this part of the Discussion was revised as “The in vitro test employs heparinized blood from vaccinated and control hens and mites feed on it through a natural or artificial membrane in an incubator at 36 °C and 75-85% relative humidity [15]. Therefore, mite mortality effects seen in in vitro feeding systems may be induced by vaccination alone or exacerbated by the high temperatures required to induce mite feeding in the chamber and the presence of anticoagulants in the blood [22, 38]. The use of the on-hen feeding device allows mites to feed directly on the hen which circumvents many of these limitations with the in vitro feeding device and is more likely to present a true reflection of vaccine effects.”
Reviewer 2 Report
The authors assessed the efficiency of Deg-AKR as a vaccine antigen to control poultry red mites. Then, the authors observed the significant reduction in oviposition of poultry red mites that fed blood from hens immunized with recombinant Deg-AKR, suggesting that Deg-AKR could be a vaccine candidate for the control of poultry red mites.
General comments
AKR/SUB is an important molecule related to several biological processes of harmful ectoparasites, and in this study, the reduction in oviposition of poultry red mites that fed blood from immunized hens was observed. Therefore, Deg-AKR could be a vaccine candidate to control the infestation of poultry red mites. Although this study was well-designed and the manuscript was constructively written, this reviewer still has some concerns as below,
1) About the feeding assay
It is difficult to understand the difference between in vitro feeding assay and on-hen feeding assay. The authors should explain how to perform the assays in more details. In this study, no difference in the mortality between fed and control groups was observed, whereas the difference was observed in the previous study. To discuss the difference in the mortality, it’s necessary to clearly understand the differences in the methods between in vitro feeding assay and on-hen feeding assay.
Were the feeding devices fixed on the thighs of immunized hens? How long did the poultry red mites feed blood? Was it independently performed four times, using the fresh (different) mites in each assay?
2) Assessment of vaccine efficacy
The authors counted the number of fed, unfed, and partially fed adults and protonymphs per hen. Why did the authors exclude deutonymphs?
3) About the expression of Deg-AKR
Which tissues is Deg-AKR expressed in? Also, which life-stages is Deg-AKR expressed in?
Other comments
Although the authors summarized the efficiency of Deg-AKR as a vaccine antigen in Table 1, the authors should address the data of other results, such as the mortality (adult, (deutonymphs?) and protonymphs) and hatching rates, depicted in Fig. 2.
Author Response
Reviewer 2
The authors assessed the efficiency of Deg-AKR as a vaccine antigen to control poultry red mites. Then, the authors observed the significant reduction in oviposition of poultry red mites that fed blood from hens immunized with recombinant Deg-AKR, suggesting that Deg-AKR could be a vaccine candidate for the control of poultry red mites.
General comments
AKR/SUB is an important molecule related to several biological processes of harmful ectoparasites, and in this study, the reduction in oviposition of poultry red mites that fed blood from immunized hens was observed. Therefore, Deg-AKR could be a vaccine candidate to control the infestation of poultry red mites. Although this study was well-designed and the manuscript was constructively written, this reviewer still has some concerns as below,
1) About the feeding assay
It is difficult to understand the difference between in vitro feeding assay and on-hen feeding assay. The authors should explain how to perform the assays in more details. In this study, no difference in the mortality between fed and control groups was observed, whereas the difference was observed in the previous study. To discuss the difference in the mortality, it’s necessary to clearly understand the differences in the methods between in vitro feeding assay and on-hen feeding assay.
Were the feeding devices fixed on the thighs of immunized hens? How long did the poultry red mites feed blood? Was it independently performed four times, using the fresh (different) mites in each assay?
Response: As described in M&M, “a feeding device was placed onto each thigh that had been the previously plucked” In response to the other question the following information was added to M&M, “Mites fed on hens for 3 hours” and “On-hen feeding experiments were repeated independently for 4 times using fresh mites on each assay on days 42, 44, 46 and 49 from first vaccination”
2) Assessment of vaccine efficacy
The authors counted the number of fed, unfed, and partially fed adults and protonymphs per hen. Why did the authors exclude deutonymphs?
Response: Deuthonymphs were not included in the infestation because they are difficult to differentiate from adult males while preparing the pouches for infestation. In response to reviewer question this following information was added to M&M.
3) About the expression of Deg-AKR
Which tissues is Deg-AKR expressed in? Also, which life-stages is Deg-AKR expressed in?
Response: This is a very interesting question, but this information is not currently available for AKR. This question should be addressed in future experiments.
Other comments
Although the authors summarized the efficiency of Deg-AKR as a vaccine antigen in Table 1, the authors should address the data of other results, such as the mortality (adult, (deutonymphs?) and protonymphs) and hatching rates, depicted in Fig. 2.
Response: In response to reviewer questions, protonymphs results were included in new Table 2. Results of oviposition in partially and fully engorged adults was included in revised R&D “However, in this experiment partially engorged adult mites did not oviposit and among fed mites only 22.4% (168 out of 749) oviposited. These facts resulted in lower number of eggs/mite (0.3 to 1.0; Table 1) when compared to previous reports (> 3.0) [32, 33]”